# Adverse or therapeutic? A mixed-methods study investigating adverse effects of Mindfulness-Based Cognitive Therapy in bipolar disorder

Imke Hanssen[1,2]☯*, Vera Scheepbouwer[1,2]☯, Marloes Huijbers[1], Eline Regeer[3], Marc Lochmann van Bennekom[2,4], Ralph Kupka[3,5], Anne Speckens[1,2]

1 Department of Psychiatry, Centre for Mindfulness, Radboud University Medical Centre, Nijmegen, The Netherlands, 2 Donders Institute for Brain, Cognition and Behaviour, Radboud University, Nijmegen, The Netherlands, 3 Altrecht institute for Mental Health Care, Outpatient Clinic For Bipolar Disorder, Utrecht, The Netherlands, 4 Pro Persona Mental Health Care, Outpatient Clinic for Bipolar Disorder, Nijmegen, The Netherlands, 5 Amsterdam UMC, Vrije Universiteit, Amsterdam Public Health Research Institute, Department of Psychiatry, Amsterdam, The Netherlands

☯ These authors contributed equally to this work.
* imke.hanssen@radboudumc.nl

**Data Availability Statement:** The data used in this study are available at the DANS Easy repository at https://doi.org/10.17026/dans-xr6-auju.

## Abstract

### Background

Mindfulness-Based Interventions (MBIs) are widely used in clinical and non-clinical populations, but little attention has been given to potential adverse effects (AEs).

### Aims

This study aimed to gain insight in the prevalence and course of AEs during Mindfulness-Based Cognitive Therapy (MBCT) for patients with bipolar disorder (BD).

### Method

The current mixed-methods study was conducted as part of a RCT on (cost-) effectiveness of MBCT in 144 patients with BD (Trial registered on 25th of April 2018, ClinicalTrials.gov, NCT03507647). During MBCT, occurrence of AEs was monitored prospectively, systematically, and actively (n = 72). Patients who reported AEs were invited for semi-structured interviews after completing MBCT (n = 29). Interviews were analysed with directed content analysis, using an existing framework by Lindahl et al.

### Results

AEs were reported by 29 patients, in seven of whom the experiences could not be attributed to MBCT during the interview. AEs were reported most frequently up to week 3 and declined afterwards. Baseline anxiety appeared to be a risk factor for developing AEs. Seven existing domains of AEs were observed: cognitive, perceptual, affective, somatic, conative, sense of self, and social. Influencing factors were subdivided into predisposing, precipitating,

**Funding:** The current study was funded by a grant from ZonMw, the Netherlands Organization for Health Research and Development (Grant Number: 843002803). The funders had no role in study design, data collection and analysis, decision to publish, or preparation of the manuscript.

**Competing interests:** AS and MH receive grants from the Netherlands Organization of Scientific Research during the conduct of this study. AS is the director of the Radboudumc Centre for Mindfulness, department of psychiatry. MH and IH are mindfulness teachers at the Radboudumc Centre for Mindfulness, department of psychiatry. This does not alter our adherence to PLOS ONE policies on sharing data and materials.

perpetuating, and mitigating factors. With hindsight, more than half of patients considered the reported AEs as therapeutic rather than harmful.

## Conclusions

Although the occurrence of AEs in MBCT for patients with BD is not rare, even in this population with severe mental illness they were not serious or had lasting bad effects. In fact, most of them were seen by the patients as being part of a therapeutic process, although some patients only experienced AEs as negative.

## Introduction

Psychotherapy research has been mainly focused on benefits, while neglecting possible (serious) adverse effects ((S)AEs) [1, 2]. Mindfulness-Based Interventions (MBIs) are widely used in both clinical and non-clinical populations, and the benefits of MBIs in psychiatric populations are well documented [3]. However, remarkably little attention has been given to potential (S)AEs following MBIs [4], even though the occurrence of (S)AEs following meditation are not uncommon. In 2017, Lindahl et al. [5] were the first to provide an elaborate overview of (S)AEs related to meditation. They interviewed 92 Western Buddhist teachers and practitioners who had experienced distressing effects from meditation and found 59 meditation-related (S)AEs across seven domains: affective, cognitive, conative, perceptual, sense of self, social, and somatic. A recent systematic review based on observational cohort studies investigating (S)AEs related to meditation (e.g. mindfulness, transcendental meditation, or zen meditation) in adults with or without a psychiatric history showed a pooled prevalence of 33%, with anxiety and depression being the most prevalent [6]. It is, however, an open question whether these results are generalizable to MBIs [7]. MBIs use a structured format, and a gradual increase in the length of meditation exercises. They include psychoeducation and inquiry, thereby allowing exchange of experiences, which might be important to prevent (S)AEs [7].

A systematic review investigating the safety of MBIs in 36 RCTs reported no difference in prevalence of (S)AEs between MBIs and control groups [4]. However, the authors suggest that attention should be paid to underreporting as only 1/5 trials structurally monitored (S)AEs. Britton et al. [8] recently published recommendations on how to define and measure meditation-related AEs in MBIs. They emphasize the importance of active monitoring of (S)AEs, using meditation-specific questionnaires as open-ended questions seem to underestimate frequencies of (S)AEs. Furthermore, they cite the Lancet Psychiatry Commission on psychological treatments [9] which recommends the exploration of (S)AEs by means of in-dept qualitative interviews.

The current study aimed to contribute to the call to actively start monitoring (S)AEs in MBIs, particularly in clinical populations [5–7]. The purpose of this mixed-methods study was threefold: 1) to investigate the prevalence of (S)AEs using meditation-specific questionnaires in patients with bipolar disorder (BD) participating in Mindfulness-Based Cognitive Therapy (MBCT); 2) to map the phenomenology of these (S)AEs onto the existing framework by Lindahl et al. [5]; and 3) to explore the course of (S)AEs and their influencing factors by in-depth interviews.

# Method

## Participants

The current study is part of an RCT on (cost-)effectiveness of MBCT + treatment as usual (TAU) compared to TAU in patients with BD [10]. Patients were recruited from outpatient clinics of seven mental health institutions across the Netherlands. The inclusion criteria were: 1) diagnosis of BD type I or II; 2) at least two lifetime depressive episodes; 3) at least one (hypo)manic or depressive episode within the year prior to baseline; 4) no severe manic symptoms at baseline (i.e. Young Mania Rating Scale (YMRS) [11] score of $\leq$ 12); and 5) no previous participation in an eight-week MBI. Written informed consent was obtained from all patients. The RCT was approved by the local medical ethics committee CMO Arnhem-Nijmegen for all participating centres (2017/3676). All procedures comply with the Helsinki Declaration of 1975, as revised in 2008.

## Intervention

The intervention was based on the original MBCT protocol for recurrent depression [12] and consisted of eight weekly group sessions of 2.5 hours and one silent day. Patients were instructed to practice at home for about 45 minutes a day with audio-guided exercises. Some adaptions to the original protocol were made in order to tailor the intervention to patients with BD. These adaptions included tailoring psychoeducation to BD, adding a partner session (session six), and instructing the teacher to add more movement exercises and repeatedly bringing focus to self-care [10]. MBCT groups consisted of eight to ten patients and were instructed by two teachers, of whom one was fully qualified according to the criteria of the UK Network for Mindfulness Based Teachers [10, 13], and one was experienced in treating patients with BD. Due to the COVID-19 pandemic, two of the twelve MBCT groups had to change to an online format in March 2020.

## Monitoring of (serious) adverse effects

During MBCT, patients completed weekly self-report questionnaires to report (S)AEs. Three psychiatrists with expertise in BD (RK, ER, and MLvB) and three clinical researchers in mindfulness (AS, MH and IH) discussed the 59 categories of meditation-related effects from the study of Lindahl et al. [5] and selected 12 most relevant to patients with BD to include in the self-report questionnaire (See S1 Table). When selecting these 12 items, it was ensured that all main categories of Lindahl et al. [5] were represented. AEs refer to any meditation related-effects that occurred during the course of MBCT, which patients indicated as having a negative valence or negative impact on daily functioning [8]. The experience was called an SAE when the outcome was death, life-threatening, hospitalization, disability, or permanent damage to conduct normal life functions or quality of life, or when it required treatment to prevent the above [14]. We also measured safety in patients undergoing TAU. At each time point of the RCT, patients were asked whether they had experienced any undesirable (medical) incident during the study period.

## Qualitative interview

All patients who reported one or more (S)AEs during MBCT were invited for an individual semi-structured interview with two researchers (IH and VS). IH is a female PhD-student, psychologist and mindfulness teacher, and VS a female resident in psychiatry who had participated in an MBSR training. Both researchers were familiar with BD, but had not been involved in the clinical care or MBCT of the participating patients. IH was responsible for the

recruitment of the RCT and therefore knew the patients. The time between completing MBCT and the qualitative interview ranged between two to 12 months (Md = 5). The following themes were discussed: 1) occurrence of AEs; 2) responses to AEs; and 3) interpretation of AEs. S2 Table provides an overview of the open-ended questions that were used during the interview. The interviews started with the question: "*What kind of adverse or unexpected experience(s) did you have that you considered to be related to MBCT?*" After that, the three above mentioned themes were discussed per AE that patients had experienced. The interviewers used open-ended questions only, factors were not specified beforehand and were therefore always reported spontaneously by patients. At the end of the interviews, when patients indicated that all AEs were discussed, the interviewers screened the self-report questionnaires to check whether that was indeed the case. One interviewer was in the lead, while the other made notes and asked clarifying questions. The duration of interviews varied between 30 and 90 minutes. Due to COVID-19 restrictions, 23/29 interviews were conducted via video-conferencing. All interviews were audio recorded and transcribed. The transcriptions and a summary were sent to the patients for verification. No repeat interviews were conducted after verification.

## Data analysis

Quantitative data were analysed using SPSS version 25.0 [15], conducting chi-square and independent t-test statistics to compare demographic and clinical variables between patients who did and did not report (S)AEs. Descriptions of the course of AEs over time were based on visual inspection.

Qualitative data were analysed by means of a directed content analysis, a method based on a deductive use of theory by validating or extending an existing theoretical framework [16]. We used the framework by Lindahl et al. [5] as a basis to identify coding categories and coded our transcripts using the predetermined codes from their codebook, which is available online [5]. Data that could not be coded were identified and analysed to determine whether it represented a new category or a subcategory of an existing code. Atlas.ti was used to analyse and classify the data [17]. To ensure reliability, transcripts were coded independently by the two interviewers (IH and VS) and codes were compared and discussed until agreement was reached. Existing coding categories and newly found codes were then discussed in the broader research team consisting of IH, VS, AS, MH, RK, ER, and MLvB.

## Results

### Prevalence and course of adverse effects

In the RCT, 144 patients were randomized to either MBCT + TAU or TAU alone (see Fig 1). Of the 72 patients randomized to MBCT + TAU, 7 (10%) patients did not start MBCT and 7 (11%) patients dropped out of MBCT, some of whom because of mania (*n* = 1) or depression (*n* = 2). The manic and one of the depressive episodes emerged before the start of MBCT, the second depressive episode emerged after start of MBCT. None of the patients who dropped-out handed in their self-report questionnaires, and therefore were excluded from further analyses. Self-report questionnaires were available for 58 (81%) patients. Of the 58 patients who handed in their self-report questionnaire, 29 (50%) patients reported one or more AEs and were invited for a semi-structured interview. Three of these (10%) were not available for an interview (one did not respond, one refused, and one died of suicide five months after MBCT —the ethical committee considered this event unrelated to MBCT). These three participants were included in the quantitative analysis. In seven interviews the experiences did not appear to be attributable to MBCT, for example pain due to a leg fracture. We concluded that in total

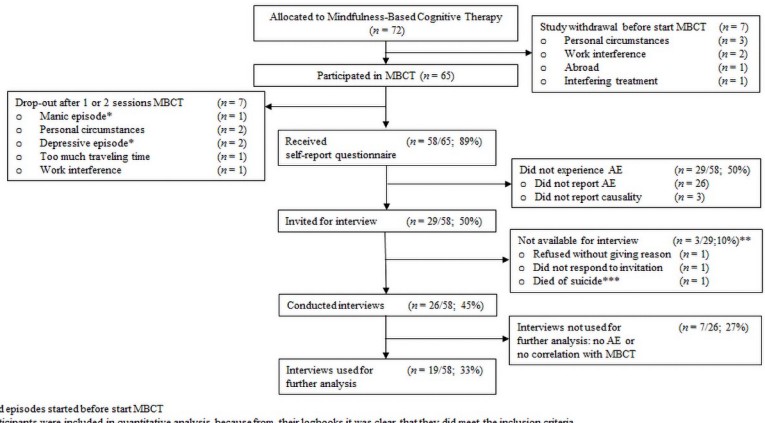

**Fig 1. Participant flowchart.**

22/58 (38%; 95% CI [0.26–0.52]) patients experienced one or more AEs due to MBCT. Patients with AEs had higher baseline levels of anxiety than those without AEs (mean difference = 5.68, 95% CI [1.405–9.959], $t(54)$ = 2.702, $d$ = 0.769). In alignment with this, four patients with AEs had comorbid panic disorder whereas only one patient without AEs had. No other demographic or clinical differences between patients with or without AEs were found.

Of the 22 patients who experienced AEs, the most frequently mentioned AEs were increase in self-related doubts ($n$ = 14; 64%; 95% CI [0.41–0.83]), uncontrollable feelings of depression ($n$ = 12; 55%; 95% CI [0.32–0.76]), anxiety or panic ($n$ = 11; 50%; 95% CI [0.28–0.72]), agitation ($n$ = 10; 46%; 95% CI [0.24–0.68]), and re-experiencing of traumatic affect ($n$ = 9; 41%; 95% CI [0.21–0.64]) (see Fig 2). Less frequently mentioned were feelings of derealization ($n$ = 8; 36%; 95% CI [0.17–0.59]), changes in trust in relation to others ($n$ = 8; 36%; 95% CI [0.17–0.59]), uncontrollable feelings of happiness/mania ($n$ = 7; 32%; 95% CI [0.14–0.55]), feelings of depersonalization ($n$ = 5; 23%; 95% CI [0.08–0.45]), strange or remarkable bodily sensations ($n$ = 4; 18%; 95% CI [0.05–0.40]), and visual hallucinations ($n$ = 2; 9%; 95% CI [0.01–0.29]).

The total number of patients who reported AEs per week declined over the course of MBCT, from 12 (21%) in week 1 to 8 (14%) in week 7 (Fig 3). In addition, the average number of reported AEs per patient per week declined, from 3.6 in week 1 to 1.5 in week 7 (Fig 4).

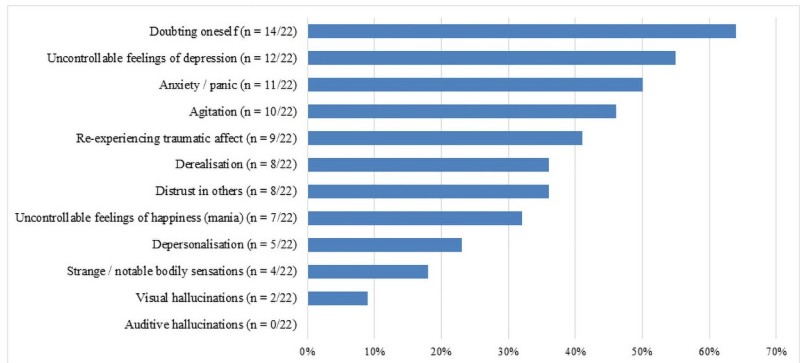

**Fig 2. Proportion of patients ($n$ = 22) reporting different types of adverse effects.**

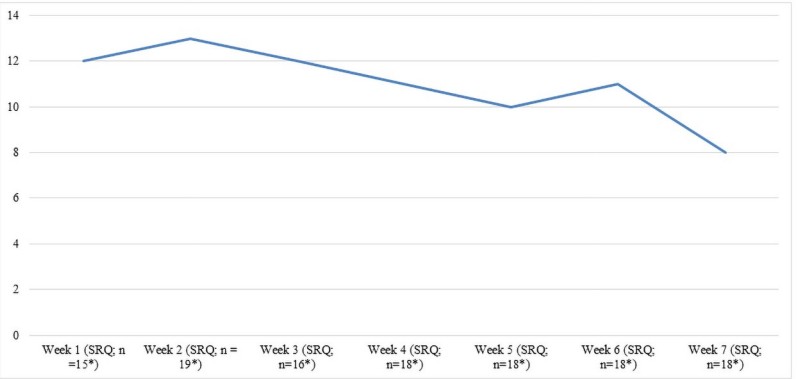

**Fig 3. Number of patients reporting adverse effects per week.**

Of the 72 patients randomized to TAU, 3 patients (4%) reported SAEs and 23 patients (32%) reported AEs. The SAEs included: surgery (*n* = 2, esophagus and unknown), and hospitalization due to severe depressive episode (*n* = 1). The AEs were divided into two main categories, namely somatic illness / physical pain (e.g. coronavirus, flu, migraine; *n* = 17), and side effects from medication (*n* = 6).

## Qualitative content analysis

Sociodemographic and clinical characteristics of the 19 patients included in the qualitative analysis are reported in Table 1.

**Replicating existing framework.** In the current study, 41 of the 59 categories of the phenomenology codebook by Lindahl et al. [5] were observed (S3 Table). *Fear, anxiety, or panic* were the most frequently mentioned AEs. Several patients experienced anxiety or panic during meditation, mostly precipitated by focusing on their breathing. When experiencing certain AEs such as *re-experiencing traumatic memories or affect* or feelings of *derealization or*

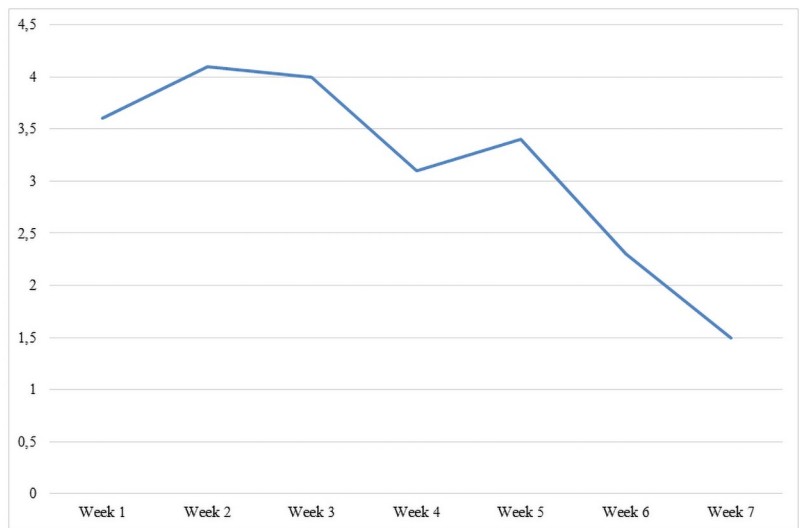

**Fig 4. Mean number of different adverse effects (AEs) reported per patient per week by patients who experienced AEs (*n* = 22).**

**Table 1. Sociodemographic and clinical characteristics of patients with bipolar disorder who participated in the semi-structured interviews about adverse effects during Mindfulness-Based Cognitive Therapy (MBCT).**

| ID | Gender | Age range | Relationship/ married | Higher education | Working | Type of BD | Duration of illness | Life-time number of mood episodes | Mood stabilizing medication | Current co-morbid anxiety disorders | YMRS score baseline[1] | IDS-C score baseline[2] | STAI-state score baseline[3] | Total number of MBCT sessions[4] | ASRM score during interview[5] | QIDS-SR score during interview[6] |
|---|---|---|---|---|---|---|---|---|---|---|---|---|---|---|---|---|
| 1 | Female | 55–60 | No | Yes | Yes | 1 | 40 | 60 | Yes | - | 0 | 14 | 40 | 9 | 5 | 6 |
| 2 | Female | 60–65 | Yes | No | No | 1 | 13 | 35 | Yes | Social phobia | 9 | 14 | 66 | 9 | 1 | 4 |
| 3 | Male | 65–70 | Yes | Yes | No | 1 | 46 | 9 | No | Specific phobia | 4 | 51 | 58 | 8 | 3 | 11 |
| 4 | Female | 35–40 | Yes | No | Yes | 2 | 22 | 6 | Yes | - | 0 | 37 | 57 | 9 | 3 | 7 |
| 5 | Female | 25–30 | No | Yes | No | 1 | 12 | 5 | Yes | Social phobia | 3 | 15 | 50 | 7 | 6 | 14 |
| 6 | Female | 40–45 | Yes | No | No | 1 | 18 | 6 | No | Specific phobia | 1 | 6 | 43 | 7 | 0 | 5 |
| 7 | Female | 35–40 | Yes | Yes | Yes | 1 | 23 | 37 | Yes | - | 0 | 6 | 42 | 9 | 0 | 2 |
| 8 | Female | 50–55 | Yes | No | Yes | 1 | 36 | 36 | Yes | - | 0 | 1 | 34 | 9 | 0 | 0 |
| 9 | Female | 30–35 | Yes | No | No | 2 | 15 | 60 | Yes | Panic disorder + OCD + PTSD | 1 | 35 | 54 | 6 | 0 | 0 |
| 10 | Female | 45–50 | No | No | No | 1 | 29 | 120 | Yes | Panic disorder + PTSD | 0 | 18 | 54 | 6 | 4 | 8 |
| 11 | Male | 50–55 | Yes | No | Yes | 2 | 19 | 9 | Yes | - | 2 | 17 | 49 | 8 | 0 | 5 |
| 12 | Male | 45–50 | No | Yes | Yes | 2 | 19 | 7 | Yes | - | 0 | 13 | 40 | 9 | 0 | 16 |
| 13 | Male | 50–55 | Yes | Yes | Yes | 1 | 31 | 3 | Yes | - | 4 | 8 | 39 | 9 | 3 | 0 |
| 14 | Female | 65–70 | No | Yes | No | 1 | 47 | 9 | No | - | 0 | 3 | 44 | 6 | 2 | 0 |
| 15 | Female | 35–40 | Yes | No | No | 2 | 29 | 42 | No | - | 8 | 6 | 46 | 8 | 5 | 8 |
| 16 | Female | 45–50 | Yes | Yes | Yes | 1 | 30 | 7 | Yes | Panic disorder | 0 | 5 | 35 | 9 | 5 | 1 |
| 17 | Female | 55–60 | Yes | No | Yes | 1 | 25 | 70 | Yes | - | 3 | 8 | 44 | 9 | 5 |  |
| 18 | Male | 35–40 | Yes | Yes | Yes | 2 | 12 | 20 | Yes | - | 4 | 36 | 49 | 6 | 0 | 14 |
| 19 | Female | 42 | Yes | Yes | No | 1 | 12 | 11 | Yes | Panic disorder + GAS | 3 | 34 | 44 | 7 | 0 | 8 |

[1] YMRS = Young Mania Rating Scale (collected at baseline as part of the ongoing RCT)

[2] IDS-C = Inventory of Depressive Symptomatology—Clinician Rated (collected at baseline as part of the ongoing RCT)

[3] STAI-state = State-Trait Anxiety Inventory (collected at baseline as part of the ongoing RCT)

[4] Including silent day

[5] Altman Self Rating Mania Scale

[6] Quick Inventory of Depressive Symptomatology—Self Rated

*depersonalization*, some patients mentioned fears about developing further mood episodes. Specific somatic sensations that patients experienced included *breathing changes*, *cardiac changes*, *gastrointestinal distress or nausea*, *tension*, and *thermal changes*. Changes in *doubt, faith, trust, or commitment* were reported by almost half or the patients, which referred to insecurities about patients' ability to learn mindfulness skills and retain a stable a mood, and doubts about whether mindfulness could be harmful in the light of the occurring AEs. *Depression* was mostly mentioned secondary to other AEs, such as *re-experiencing traumatic memories or affect* and *changes in doubt*. Furthermore, *agitation or irritability*, either secondary to other AEs or in response to specific types of practices, was frequently reported as well.

Almost all patients experienced multiple AEs at the same time. One AE often seemed to precipitate further AEs, sometimes resulting in a vicious cycle. For example, participant #2 described that she became anxious when focusing on her breathing, after which she started to doubt her capability to meditate (*"I can't do this"*). These negative thoughts precipitated several somatic sensations, such as tension and sweating. She described that by focusing on these sensations during meditation, she became even more anxious, resulting in a panic attack. Afterwards, she described that she felt depressed, which made her more prone to feeling anxious, which precipitated other panic attacks during further meditations.

The 18 categories that were not found included SAEs such as: *suicidality; delusional, irrational, or paranormal beliefs; disintegration of conceptual meaning structures*; and *change in executive functioning*. No additional codes to the phenomenology codebook were found. As no SAEs were found in this study, the text below will only mention AEs.

**Extending the existing framework.** With regard to the influencing factors, the current study observed all 26 categories that were previously found in the study by Lindahl et al. [5] (S4 Table). In addition, we categorized these influencing factors into four new coding categories, i.e. predisposing, precipitating, perpetuating, and mitigating factors. Furthermore, we investigated which consequences these influencing factors had on the AEs, as reflected in the 'consequences' domain.

*Predisposing factors.* Patients mentioned several factors that existed before the start of MBCT and that might predispose to the occurrence of AEs. The most frequently reported predisposing factor was *psychiatric history*, which included the experience of previous mood episodes, psychosis, anxiety, or hospitalization because of severe manic or depressive episodes. *Trauma history* and *mood symptoms* that were apparent before the start of MBCT were also considered important predisposing factors, as were *personality* traits such as perfectionism, performance anxiety, emotional hypersensitivity, and self-criticism. Somatic co-morbidity or *medical history*, including chronic pain, fatigue, and menopause were mentioned, as were certain interpersonal circumstances that occurred in *relationship with others*, such as death or illness of family members, marriage, or divorce. A history with similar experiences as the occurred AEs with regard to *previous meditation and yoga practice* was reported. Finally, specific *intentions, motivations, or goals*, for example expecting that MBCT would have a calming effect, were thought to predispose to AEs.

*Precipitating factors.* Overall, patients reported the *increased awareness* due to meditation practice as the most important precipitating factor. Specific *types of practice* were mentioned as triggers for certain AEs. For example, focusing on breathing precipitated feelings of anxiety and panic. Some patients mentioned the *amount of practice* as precipitating, referring to practices of 40 minutes, which induced the above mentioned vicious cycle of AEs more easily. *Surroundings* were mentioned, referring to sounds that interrupted their focus of attention, resulting in AEs such as agitation or doubt. The *relationship to teacher* was reported, for example if the teacher repeatedly used the word 'breath', bringing the attention back to the precipitating factor of focus on breathing. Furthermore, the intonation of the voice of the teacher

during meditation was mentioned, inducing a deep internal focus which triggered trauma-related memories. Wanting to please the teacher was mentioned, for example that some patients felt they should not move when feeling uncomfortable during meditation. *Relationships within meditation community*, including anxiety for new group members and hearing others' stories about severe manic episodes, were mentioned as well. Eventually, *relationships beyond meditation community* at work or within the family could be precipitants when they were stressful to patients.

*Perpetuating factors.* Patients mentioned several perpetuating factors, which refers to specific responses that were considered to be unhelpful because they had a negative effect on the number or intensity of AEs. Some *maladaptive (automatic) coping strategies* of patients themselves, such as trying to continue the meditation exercises despite the occurrence of AEs, were considered as perpetuating. Several patients reported their *relationship to the teacher* as perpetuating, because they felt the teacher did not take the AE seriously or did not ask about AEs while discussing homework. Some specific unhelpful reactions of the teacher were mentioned, such as suggesting that they should "stay with the experience" or adopt a non-judgmental and accepting attitude towards the AE. With regard to their *relationships within meditation community*, some patients mentioned that they were reluctant to move or stop practicing when they experienced AEs, because they did not want to bother their peers or were afraid of disapproval.

*Mitigating factors.* Responses were considered mitigating when the AEs reduced in number or intensity, or completely disappeared. *Mindfulness skills* that were used during meditation practice to anchor in the present moment were considered helpful, including adapting a more accepting, compassionate or soothing attitude towards the experience, and investigating or observing the experience from a distance (i.e. decentering). For example, patient #5 stated: "*I was actually just trying to observe it: just observe it and you'll see that you don't have to be afraid of it.*" Certain *grounding activities that were considered helpful* included opening eyes, changing posture, feeling their feet on the floor, focusing on the breath, or touching a nearby object. Patients also mentioned different approaches, such as stopping their practice or redirecting attention when the AEs became too intense (i.e. doing dishes), practicing less frequent or shorter, or writing down their experiences. Furthermore, patients found it helpful to share their experience of AEs with the *teacher*, and with *people within and beyond the meditation community*. Several reactions from the teacher were mentioned, such as hearing that their experiences were not uncommon and suggesting alternative ways to practice (e.g. focusing on the feet instead of the breath). Sharing their experiences with people within the meditation community made patients feel accepted, relieving some of the impact and shame following AEs. They described the meditation group as empathic, open and safe. Some patients also found it helpful to mention AEs to their *mental health care professional*. One participant started a trajectory of EMDR after experiencing trauma-related memories during the MBCT that had been suppressed. A few patients used *medication* (e.g. benzodiazepines) to alleviate the intensity of AEs (in cases of anxiety and panic).

*Consequences.* First of all, some patients believed that the AEs would persist or even worsen over time and tended to avoid practicing mindfulness altogether in order to prevent this. However, more than half of patients indicated that the AEs had been negative and challenging at first, but that with hindsight they regarded them as part of a therapeutic process (*n* = 11; 58%). The AEs evoked fear at first, for example fear of developing mood episodes, but they realised that the AEs did not provoke mood episodes. Patients mentioned that due to the AEs, they had been able to learn and develop specific mindfulness skills, such as acceptance, allowing, decentering, compassion and self-care, which might be helpful to deal with other difficult situations. For example, participant #19 stated: "*But with what I know now, that it worked as a magnifying*

*glass, but also gave me some very good tools. I don't know if I could have reached this point without going through that* (AE: panic)." Some patients experienced AEs as merely having negative impact, because they did not feel that they had learned anything from it (*n* = 8; 42%).

## Discussion

### Summary and comparison with literature

The current mixed-methods study actively and systematically monitored the occurrence of AEs during MBCT for patients with BD. The prevalence of patients who reported one or more AEs during the eight-week MBCT was 50%. For seven patients their experiences appeared not to be attributable to MBCT after all, resulting in a corrected prevalence of 38%. Directed content analysis of semi-structured interviews with patients who experienced AEs observed seven domains of AEs as outlined in the framework by Lindahl et al. [5]: cognitive, perceptual, affective, somatic, conative, sense of self, and social. All categories of influencing factors were observed in the current study, which were categorized into four new coding categories, i.e. predisposing, precipitating, perpetuating, and mitigating factors.

The prevalence of AEs in the current study (38%) is comparable with the estimated prevalence of AEs during meditation (33%) in a recent systematic review in adults with or without a psychiatric history [6] and with a prevalence of 32% in a large population-based sample who had previous meditation experience in the United States [18]. The number of patients reporting AEs and the different types of AEs appeared to decline during the course of MBCT. From the start of MBCT, patients learn to intentionally direct their attention to their present moment experience, even if that experience is unpleasant and difficult [12]. The current study shows that increased awareness was mentioned as a precipitating factor for AEs. Indeed, higher levels of awareness have been associated with increased levels of depression and anxiety [19]. During the course of MBCT patients gradually learn to develop acceptance and self-kindness, which have been shown to reduce psychological distress [12, 19, 20]. This might explain why the number of AEs reported by patients starts to decline after three weeks. However, another explanation of this decline may be that even though patients still experienced AEs during the course of MBCT, they may have become reluctant to report this on the self-report questionnaires.

Although the occurrence of AEs is not rare, even in this population with severe mental illness, we did not find any SAEs. All the seven larger domains of AEs as previously defined by Lindahl et al. [5] were observed in the current study, but we did not find SAEs such as suicidality, disintegration of conceptual meaning structures, delusional or irrational beliefs, or changes in executive functioning. Furthermore, we did not find any lasting bad effects, which is in contrast with Britton et al. [8] who found that 15% of participants still experienced lasting bad effects for longer than one week after the (S)AEs following MBCT had occurred. One explanation for this difference might be that this population of patients with BD had a long treatment history in which they had possibly become familiar with adverse emotional experiences such as panic attacks, re-experiencing of traumatic memories, depersonalization, and with ways to cope with them. Furthermore, by actively monitoring (S)AEs is it possible that patients were inclined to share these challenging experiences sooner, after which the teacher was able to provide them with tools to mitigate these (S)AEs.

More than half of patients described that the AEs were negative and challenging at first, but helped them in their therapeutic process, because these difficult experiences had helped them to learn and develop new skills that were helpful in the long run. This finding is in line with the study by Goldberg et al. [18] who found that around 88% of participants was feeling glad to have practiced meditation even though they had experienced meditation-related AEs.

## Clinical implications

The influencing factors that were mentioned by patients were categorized into predisposing, precipitating, perpetuating, and mitigating factors, and provide important clinical implications to prevent AEs in MBIs. The predisposing factors can help clinicians and teachers to assess whether MBCT is suitable for a particular patient at a particular time. In case of a risk factor such as high baseline anxiety, one might be more careful [21]. Teachers and clinicians should actively inquire about anxiety symptoms and co-morbid anxiety disorders at baseline, and perhaps adapt their guidance, for example by closely monitoring patients and providing them with tools to prevent or mitigate AEs. Alternatively, anxiety symptoms could be treated first, for example using CBT with exposure.

Both precipitating and perpetuating factors provide insight into ways to improve MBIs, for example by managing expectations and inform patients about the possibility that they might become more aware of difficult emotions or that unexpected, negative effects might occur. Informing patients in advance may help to come to a balanced and ideally shared decision, with the mindfulness teacher on whether to stay with the practice (with perhaps some adaptations), or to consider other treatment alternatives. During practice, it is important to reassure patients that they can always choose to take a different posture, to open their eyes, or to take a break. Normalizing experiences and reassuring patients who experience difficult emotions seems helpful. This emphasizes the importance of actively asking about negative or unexpected experiences during the inquiry of exercises and homework.

The mitigating factors may also be helpful in dealing with AEs. For example, teachers could encourage patients to stay within their 'window of tolerance' [22] and to be aware of situations in which 'just staying with the experience' might not be the best instruction. In some cases it might be best to invite patients to adapt or change the (type of) exercise. For people who have a psychiatric history it seems important to offer MBIs embedded in a mental healthcare setting. In clinical populations, the occurrence of AEs might precipitate a relapse of psychiatric symptoms (e.g. depression). During the current study, MBCT was offered by mental health professionals, allowing an evaluation of possible first warning signs of a psychiatric disorder and prevention of deterioration by adding other treatments.

## Strengths and limitations

A major strength of the current prospective mixed-methods study is the active and systematic monitoring of AEs during MBCT in patients with severe mental illness. Data were analysed within an existing theoretical framework. All patients who reported one or more AEs were invited for semi-structured interviews. On the other hand, active monitoring could also result in overreporting of AEs. In fact, the semi-structured interviews indeed revealed that some AEs were not attributable to MBCT. Overreporting of AEs also became apparent in a recent study exploring the prevalence and severity of (S)AEs following MBCT in teachers and students, showing that 67% of participants reported unpleasant experiences (e.g. unpleasant emotions, thoughts, and sensations), but only 2–7% were considered harmful [23]. However, we cannot rule out the possibility of underreporting either. Patients who dropped out of MBCT or did not hand in the self-report questionnaires were not approached, while SAEs might be overrepresented among them. Teachers and clinicians should actively inquire about (S)AEs when patients drop out. Carefully composed questionnaires, such as ours, with both closed and open-ended questions to prevent over- or underreporting, are needed. Another possible limitation of the study is the sample size, which might not be large enough to identify (S)AEs that occur less frequently [24]. More studies systematically investigating (S)AEs in MBIs are needed, so data can be pooled and estimates of the prevalence of (S)AEs can be refined.

## Supporting information

**S1 Table. Self-report questionnaire to monitor the occurrence of adverse effects (AEs) during MBCT.**
(DOCX)

**S2 Table. Topic-list used during interviews.**
(DOCX)

**S3 Table. Phenomenology codebook: Number and proportions of predetermined (Lindahl et al., 2017) found adverse events in interviews during Mindfulness-Based Cognitive Therapy in patients with bipolar disorder ($n$ = 19).**
(DOCX)

**S4 Table. Influencing factors codebook: Percentage of predetermined (Lindahl et al. 2017) and newly found influencing factors of adverse effects as reported by patients with bipolar disorder during Mindfulness-Based Cognitive Therapy ($n$ = 19), illustrated with quotes.**
(DOCX)

**S1 Checklist.**
(PDF)

**S1 Study protocol.**
(PDF)

## Acknowledgments

The authors would like to thank the mindfulness teachers and patients who participated in this study.

## Author Contributions

**Conceptualization:** Imke Hanssen, Marloes Huijbers, Eline Regeer, Marc Lochmann van Bennekom, Ralph Kupka, Anne Speckens.

**Data curation:** Imke Hanssen, Vera Scheepbouwer, Marloes Huijbers.

**Formal analysis:** Imke Hanssen, Vera Scheepbouwer.

**Funding acquisition:** Marloes Huijbers, Anne Speckens.

**Investigation:** Imke Hanssen, Vera Scheepbouwer.

**Methodology:** Imke Hanssen, Marloes Huijbers, Eline Regeer, Marc Lochmann van Bennekom, Ralph Kupka, Anne Speckens.

**Supervision:** Marloes Huijbers, Eline Regeer, Marc Lochmann van Bennekom, Ralph Kupka, Anne Speckens.

**Writing – original draft:** Imke Hanssen, Vera Scheepbouwer.

**Writing – review & editing:** Imke Hanssen, Marloes Huijbers, Eline Regeer, Marc Lochmann van Bennekom, Ralph Kupka, Anne Speckens.

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
