## [Decision Letter · Decision Letter 0]

16 Jul 2021

PONE-D-21-18959

Adverse or therapeutic? A mixed-methods study investigating adverse effects of Mindfulness-Based Cognitive Therapy in bipolar disorder

PLOS ONE

Dear Dr. Hanssen,

Thank you for submitting your manuscript to PLOS ONE. After careful consideration, we feel that it has merit but does not fully meet PLOS ONE’s publication criteria as it currently stands. Therefore, we invite you to submit a revised version of the manuscript that addresses the points raised during the review process.

We obtained statistical and content area expertise from the referees. Both referees appreciated the timeliness of your manuscript. I believe all of their comments are valuable and worth integrating into your revisions.

We look forward to receiving your revised manuscript.

Kind regards,

Ethan Moitra

Academic Editor

PLOS ONE

Brown University

Journal Requirements:

"AS and MH receive grants from the Netherlands Organization of Scientific Research during the conduct of this study. AS is the director of the Radboudumc Centre for Mindfulness, department of psychiatry. MH and IH are mindfulness teachers at the Radboudumc Centre for Mindfulness, department of psychiatry."

Reviewers' comments:

Reviewer's Responses to Questions

**Comments to the Author**

1. Is the manuscript technically sound, and do the data support the conclusions?

Reviewer #1: Yes

Reviewer #2: Partly

2. Has the statistical analysis been performed appropriately and rigorously? 

Reviewer #1: Yes

Reviewer #2: N/A

3. Have the authors made all data underlying the findings in their manuscript fully available?

Reviewer #1: Yes

Reviewer #2: No

4. Is the manuscript presented in an intelligible fashion and written in standard English?

Reviewer #1: Yes

Reviewer #2: Yes

5. Review Comments to the Author

Reviewer #1: This is generally a well written paper. I will focus on quantitative methods and reporting

Major

1) I imagine that SAEs are not relevant to the control group? In any other intervention (think vaccination for example) we would compare SAEs in the two groups. Can the authors make is explicitly clear, why this design (looking only into the intervention group) is appropriate both in the abstract and in the methods section?

2) Characteristics associated with SAEs are not examined, in say a multiple logistic regression (or even a zero-inflated Poisson regression to capture multiple SAEs).

3) When reporting focus on association sizes and their confidence intervals - move away from p-values a bit.

Minor

1) Prevalence and other estimates form the sample needs to be reported with levels of uncertainty (confidence intervals). see exact, wald, wilson, agresti, and jeffreys binomial confidence intervals, and choose one.

Reviewer #2: *This review is also included as a separate document, attached*

Thank you for the opportunity to review this manuscript of a well-designed study examining AEs in an impressive sample of patients diagnosed with bipolar disorder undergoing MBCT. There is much to recommend this article, which emerges at a time when there is growing interest in AEs in mindfulness based programs. The monitoring of adverse events was well thought out and well executed. With some important revisions pertaining to interpretation, clarity, and description of the findings, I believe this would make a valuable addition to the literature on AEs in mindfulness programs. Below are some general comments, followed by specific items to address.

As I have mentioned, certain components of this manuscript would benefit from revision. Features of the methodology would benefit from more detailed explanation, and aspects of the study methods suggest (mostly minor) revisions to the authors’ interpretations of its findings. Notably, the qualitative analysis framework should be explained in greater detail, including the specific framework used and the steps employed in that framework. Elements of the qualitative data analysis and categorization were unclear to this reviewer, including the designation of SAEs vs. AEs, how the relevant findings emerged. Some examples: were mitigating factors queried specifically, or spontaneously volunteered by the interviewers? Were the categories etic or emic? Similarly, how were some of the predisposing factors determined? If psychiatric history was only gleaned from spontaneous responses during the interview—the interview, as currently described, only lists 3 topics which do not include psychiatric history—then estimates of psychiatric are likely to be inaccurate. Was this information gathered in any other way? If not, this should be stated clearly, and listed as a limitation of the study and its interpretation. It was also not clear why those who had dropped were excluded from consideration as individuals who may have had potential SAEs. For instance, although this is listed briefly as a limitation in the study, it seems that if a suicide occurred for one of the patients, it would be premature to foreclose the possibility of it being an AE in the study. A number of similar concerns are listed point by point below. A broader methodological concern is that the authors describe this research as “confirming” Lindahl et al.’s (2017) categories, but the study does not seem optimally designed to confirm, rather than to observe or apply, Lindahl, et al.’s framework (e.g., given the smaller number of people reporting AEs in the current study, and the difference between Buddhist meditators and MBCT patients). This does not detract from the merit of the study, but does make this reviewer hesitant about interpreting this research as “confirmatory.”

This paper should also enter into closer conversation with some recent literature. Namely, Britton, et al.’s (2021) paper was likely published after submission of the present manuscript, but provides valuable guidelines for measuring harms, severity, duration, etc. Another recent paper that the authors may not have encountered at the time of submission is Goldberg, et al. (2021), which also addresses the prevalence of adverse experiences in mindfulness. Other research that went unmentioned but is probably relevant is Schlosser, et al. (Schlosser et al., 2019). Just as the Lindahl, et al. paper provided a useful typology of meditation-related challenges, the recent Britton, et al. (2021) paper provides guidelines (which the authors do not have to follow, but should be aware of/in dialogue with) for measuring harms in MBPs. It is to the current authors’ credit that they have done a very rigorous job in measuring harms, so this should not present much of a challenge. Another question related to the background literature pertains to the role of Lindahl, et al.’s work in this manuscript. The authors do impressive work applying the domains observed by Lindahl, et al., but should also mention early on that the preceding work was done in long-term Western Buddhist meditators, and generalization to MBIs should be undertaken with caution.

Abstract

- “Interviews were analysed with directed content analysis, using an existing framework [1].”

Please name the framework

- “The seven existing domains of AEs were confirmed: cognitive, perceptual, affective, somatic, conative, sense of self, and social.”

Take out “the”, and I suggest changing language from “confirmed” to “observed”, see below.

Introduction

- “Mindfulness-Based Interventions (MBIs) are widely used in both clinical and non-clinical populations, and although the benefits of MBIs in psychiatric populations are well documented [4], the same applies to MBIs.”

Grammatically confusing. What is “the same” that applies to MBIs?

- As mentioned earlier, Britton, et al. 2021 should be cited, as that article outlines a methodological approach and has described base rates for AEs. Also Goldberg, et al. 2021 will be relevant.

- I suggest caution in generalizing Lindahl, et al.’s study to MBIs, as that research was done with long-term Buddhist meditators. This article mentions this as an aside at the end, but it should be clear from the outset.

Method

- The authors mention that this research relied on an adapted version of MBCT. Reasons and implications of adaptation should be stated (perhaps in discussion is OK).

- Weekly completion of questionnaires is a definite strength. However, if only 12 AEs that were a-priori deemed most relevant to BD were indicated, this may also leave important AEs out. AEs may have as much to do with the intervention as with the population, so making the items only focused on the specifics of the population may miss important characteristics. By way of analogy, an (S)AE of aspirin is Reye’s syndrome. However, if probing AEs of aspirin among low back pain sufferers, only asking about low back-relevant AEs would miss this serious AE. To be clear, the current questions seem appropriate and—if selecting 12—the items used appear to be good choices even if they do leave out (for example) change in social relationships.

- Small concern: IH knew the patients and was responsible for recruiting them. Was there any concern that patients may under-report AEs to the person who recruited them, especially if she is a junior researcher?

- The authors discuss the construal of AEs as ultimately positive in many cases. For reference in the varied interpretation of meditation-related AEs, and some of the context around positive vs. negative interpretation of AEs as “progress”-related, they may also find a recent paper by Lindahl, et al. (2020) relevant.

- Was any kind of intent to treat approach used with drop-outs? Was there any indication of why people dropped out and whether this may have been related to AEs? Also, it’s not clear to what extent drop-outs are taken into account in the current analyses. A problem sentence is: “Of these, 29 (50%) patients reported one or more AEs and were invited for a semi-structured interview…”

“of these” is confusing, since prior to this sentence a general sample is listed followed by number drop-outs. So is this referring to all participants prior to drop-out, or after excluding drop-outs?

- The description of qualitative analyses is quite sparse. Are there any data on initial rates of agreement between coders? What was the “existing framework” for directed content analysis? Etc.

- Was chronicity/duration of AEs monitored? If so, how? The authors indicate later in the paper that enduring AEs were not observed, but it is not clear how they are able to make such a statement, especially if AEs were not monitored after end of trial.

Results

- “In the current study, 41 of the 59 categories of the phenomenology codebook by Lindahl et al. [1] were confirmed (Supplementary Table 3).”

I suggest stating “observed” instead of “confirmed”, since the design of the study is not well suited to confirm or disconfirm these categories (i.e., the authors did not produce an independent coding system and find it to accord with the Lindahl, et al. themes, nor did they sample an adequate number of participants (22 vs. approximately 60 in the Lindahl et al. study). To be clear this is not a problem with the study or its methods, which are sound. Rather, this is just about what the study is and isn’t positioned to do. I therefore suggest “observed.”

- “Both the number of patients who reported AEs and the average number of different AEs per patient declined over the course of MBCT, from 12 (21%) in week 1 to 8 (14%) in week 7 and from 157 3.6 in week 1 to and 1.5 in week 7, respectively (see Fig 3a and Fig 3b).”

This is a bit confusing – are these total patients or averaged numbers of AEs? I am assuming that it’s patients followed by averages, but please make this clear.

- “Changes in doubt, faith, trust, or commitment were frequently reported as well, which referred to insecurities about patients’ ability to learn mindfulness skills and retain a stable a mood, and doubts about 172 whether mindfulness could be harmful in the light of the occurring AEs.”

What is “frequently”?

- “As no SAEs were found in this study, the text below will only mention AEs.”

- The authors do not explain how “seriousness” was determined, a very important omission to correct. This explanation would naturally fall in the “monitoring serious adverse effects” section, but that section only includes the questionnaire and interview approach.

- Please define each of the 4 new categories for influencing factors. I.e., the authors write “Patients mentioned several factors that existed before the start of MBCT and that might predispose to the occurrence of AEs.” Are these factors deemed by the patients, or by the researchers, to be predisposing? And if by the researchers, based on what criteria? How were these predisposing factors (i.e., personality traits) identified – were they spontaneously reported by participants, or queried by interviewers?

- Please explain the categorization of “trying to continue the meditation exercises despite the occurrence of AEs” as a maladaptive (automatic) coping strategy, given that instructions to “meditate through it” are a conspicuous concern in popular meditation instruction. If this was in line with existing instruction and literature on meditation, calling it “automatic” localizes this issue to the patient, rather than placing appropriate responsibility with instructors (perhaps not those in the present study) who repeat this instruction.

- “Mindfulness skills” as a mitigating factor is not an immediately clear category – please provide a definition/explication, including how the category was generated. Were these described as “mindfulness skills” by participants, or categorized as such by the researchers? How are these different from “ground activities” (which include focusing on the breath) for instance?

- It will be important to resolve the following concern: The article initially states that most patients interpreted the AEs to be ultimately therapeutic or positive. However, the Consequences section begins by writing “First of all, some patients believed that the AEs would persist or even worsen over time and tended to avoid practicing mindfulness altogether in order to prevent this.” Or at the end of that paragraph, “Some patients experienced AEs as merely having negative impact, because they did not feel that they had learned anything from it.” Given the small number of participants, this is non-trivial and would seem to indicate that the earlier statement be revised. One possible revision would be to add something like “although some only experienced AEs as negative.” With monitoring adverse effects, it is important not to elide the experiences of those patients. It is also important to bear in mind that the full extent of distress may not be observed within the time window of the study. For example, when comparing AEs that did vs. did not reduce over time (Supplementary Table 4), the number that do not decrease outnumber those that did. Likewise, the “majority” who interpreted the AEs as part of a growth process was 11 vs. 8 who did not, which I is a VERY SLIM majority. Since an AE like depersonalization can seem fine within the time frame of an 8-week MBCT treatment but, without improvement, can become highly distressing after 3 months, it is inappropriate to characterize these as mostly positive overall.

- The Consequences section again mentions specific “mindfulness skills” which makes me wonder whether these were queried in the interview, or whether they were ultimately categorized as such by the researchers.

- Given the spread of those who did and did not find AEs to ultimately be beneficial, it might be useful to see the numbers behind these differences.

- “During the course of MBCT patients gradually learn to develop acceptance and self-kindness, which have been shown to reduce psychological distress [10, 15, 16]. This might explain why the number of AEs reported by patients starts to decline after three weeks. These results suggest that allowing difficult or adverse emotions can result in a reduction of associated fears and habituation. This is in accordance with exposure-based therapies, where a temporary increase of anxiety is considered a necessary vehicle to change patients’ beliefs and emotions [7]” This may be overstating the case, and I would suggest caution in making such an interpretation given the kind of—and amount of—data here. For instance, an alternative explanation is that, as with many adverse experiences, impacts are shorter-term, likelier to occur in tandem with any novel experience, and that they recover on their own because people are by and large resilient. Decline in reporting could be due to Hawthorne effects, as patients who report an AE 3 weeks in a row may be reluctant/tired/habituated to it on week 4 even though they still experience it. It is thus a strength of the study that these patients were subsequently interviewed (and yet the interviews did not suggest that decrease was the majority experience). Finally, research suggests that early losses in treatment are not good predictors of longer-term treatment gains (e.g. “it will get worse before it gets better”)—this is a common assumption but is not well supported (Flückiger et al., 2013; Koffmann, 2018, 2020; Lutz et al., 2013).

- “In the current study, with a vulnerable population of patients with BD, MBCT seemed reasonably safe.”

This is a value judgment. See paper by Britton, et al. (2021) for discussing safety.

- Given that one patient died by suicide (a low base rate event), I am concerned about the authors’ justification in saying they did not observe suicidality as an AE among patients.

- Given the low base rates of harms in general, it is not justified based on lack of observation in this trial to say that these do not occur.

- “Both precipitating and perpetuating factors provide insight into ways to improve MBIs, for example by managing expectations and inform patients about the possibility that symptoms may initially get worse” See earlier comment that “things get worse before they get better” is not well borne out in psychotherapy, and that early losses do not predict later gains.

- “Informing patients in advance may prevent them from dropping out due to these AEs.” Why is it inappropriate to drop out due to AEs? This seems like it would be a reasonable strategy with many other therapies. It also stands in contrast with the researchers’ observations that persisting with the practice is a potential exacerbating factor. I suggest changing this language to indicate that patients should received informed guidance about their options to drop out, switch therapies, etc. in the event of AEs.

- The authors state that long-term AEs were not observed, but since all AEs did not disappear at end of study, how can they know?

- Not sure why the patterns described in strengths and limitations are deemed “over”-reporting. It seems appropriate to query AEs and have a system for determining whether these were harmful, but the methods are well designed to correct for any over-estimation due to the sensitivity of the instruments.

- A limitation is that drops due to AEs were not recorded.

References

Britton, W. B., Lindahl, J. R., Cooper, D. J., Canby, N. K., & Palitsky, R. (2021). Defining and Measuring Meditation-Related Adverse Effects in Mindfulness-Based Programs. Clinical Psychological Science, 2167702621996340. https://doi.org/10.1177/2167702621996340

Flückiger, C., Holtforth, M. G., Znoj, H. J., Caspar, F., & Wampold, B. E. (2013). Is the relation between early post-session reports and treatment outcome an epiphenomenon of intake distress and early response? A multi-predictor analysis in outpatient psychotherapy. Psychotherapy Research, 23(1), 1–13. https://doi.org/10.1080/10503307.2012.693773

Goldberg, S. B., Lam, S. U., Britton, W. B., & Davidson, R. J. (2021). Prevalence of meditation-related adverse effects in a population-based sample in the United States. Psychotherapy Research: Journal of the Society for Psychotherapy Research, 1–15. https://doi.org/10.1080/10503307.2021.1933646

Koffmann, A. (2018). Has growth mixture modeling improved our understanding of how early change predicts psychotherapy outcome? Psychotherapy Research, 28(6), 829–841. https://doi.org/10.1080/10503307.2017.1294771

Koffmann, A. (2020). Early trajectory features and the course of psychotherapy. Psychotherapy Research, 30(1), 1–12. https://doi.org/10.1080/10503307.2018.1506950

Lindahl, J. R., Cooper, D. J., Fisher, N. E., Kirmayer, L. J., & Britton, W. B. (2020). Progress or Pathology? Differential Diagnosis and Intervention Criteria for Meditation-Related Challenges: Perspectives From Buddhist Meditation Teachers and Practitioners. Frontiers in Psychology, 0. https://doi.org/10.3389/fpsyg.2020.01905

Lutz, W., Ehrlich, T., Rubel, J., Hallwachs, N., Röttger, M.-A., Jorasz, C., Mocanu, S., Vocks, S., Schulte, D., & Tschitsaz-Stucki, A. (2013). The ups and downs of psychotherapy: Sudden gains and sudden losses identified with session reports. Psychotherapy Research, 23(1), 14–24. https://doi.org/10.1080/10503307.2012.693837

Schlosser, M., Sparby, T., Vörös, S., Jones, R., & Marchant, N. L. (2019). Unpleasant meditation-related experiences in regular meditators: Prevalence, predictors, and conceptual considerations. PLOS ONE, 14(5), e0216643. https://doi.org/10.1371/journal.pone.0216643

6. PLOS authors have the option to publish the peer review history of their article (what does this mean?). If published, this will include your full peer review and any attached files.

Reviewer #1: No

Reviewer #2: No

---

## [Author Response · Author response to Decision Letter 0]

19 Aug 2021

[Reviewer #1] 

We thank the reviewer for these comments. We did our best to incorporate all of the remarks. The mentioned pages and lines refer to the highlighted manuscript file. 

Major

1) I imagine that SAEs are not relevant to the control group? In any other intervention (think vaccination for example) we would compare SAEs in the two groups. Can the authors make is explicitly clear, why this design (looking only into the intervention group) is appropriate both in the abstract and in the methods section?

[Authors] We agree with the reviewer that ideally, we should have investigated the occurrence of (S)AEs in both the control and intervention group. During the RCT, we did measure safety in both groups at each time point of the study, in the sense that we asked patients if they had experienced any undesirable (medical) incidents during the study period, without it having to be related to the study itself. We included this data in the manuscript. 

In the methods section under ‘monitoring of (serious) adverse effect we included: “We also measured safety in patients undergoing TAU. At each time point of the RCT, patients were asked whether they had experienced any undesirable (medical) incidents during the study period.” (page 5, lines 121-123). 

In the results section we included: “Of the 72 patients randomized to TAU, 3 patients (4%) reported SAEs and 23 patients (32%) reported AEs. The SAEs included: surgery (n = 2, esophagus and unknown), and hospitalization due to severe depressive episode (n = 1). The AEs were divided into two main categories, namely somatic illness / physical pain (e.g. coronavirus, flu, migraine; n = 17), and side effects from medication (n = 6).” (page 8, lines 208-212). 

2) Characteristics associated with SAEs are not examined, in say a multiple logistic regression (or even a zero-inflated Poisson regression to capture multiple SAEs).

[Authors] We did explore whether baseline characteristics (e.g. age, gender, co-morbid psychiatric disorders, number of previous mood episodes, duration of illness, baseline depression, etc.) differed between patients who did and did not report (S)AEs. We did not find any differences, except for baseline anxiety (see page 7, line 178). This is an important finding, and we elaborated on this in the discussions section (page 14, line 402-407). 

3) When reporting focus on association sizes and their confidence intervals - move away from p-values a bit.

[Authors] We replaced the p-value mentioned in text by the association size and confidence interval: “Patients with AEs had higher baseline levels of anxiety than those without AEs (mean difference = 5.68, 95% CI [1.405 – 9.959], t(54) = 2.702, d = 0.769).” (page 7, line 178-180).

Minor

1) Prevalence and other estimates from the sample needs to be reported with levels of uncertainty (confidence intervals). see exact, wald, wilson, agresti, and jeffreys binomial confidence intervals, and choose one.

[Authors] We calculated the confidence levels of the prevalence and added this to the manuscript: “We concluded that in total 22/58 (38%; 95% CI [0.26 – 0.5]) patients experienced one or more AEs due to MBCT.” (page 6, line 177-178). 

Furthermore, we included the CI of the prevalence of different types of AEs:

“Of the 22 patients who experienced AEs, the most frequently mentioned AEs were increase in self-related doubts (n = 14; 64%; 95% CI [0.41 – 0.83]), uncontrollable feelings of depression (n = 12; 55%; 95% CI [0.32 – 0.76]), anxiety or panic (n = 11; 50%; 95% CI [0.28 – 0.72]), agitation (n = 10; 46%; 95% CI [0.24 – 0.68]), and re-experiencing of traumatic affect (n = 9; 41%; 95% CI [0.21 – 0.64]) (see Fig 2). Less frequently mentioned were feelings of derealization (n = 8; 36%; 95% CI [0.17 – 0.59]), changes in trust in relation to others (n = 8; 36%; 95% CI [0.17 – 0.59]), uncontrollable feelings of happiness/mania (n = 7; 32%; 95% CI [0.14 – 0.55]), feelings of depersonalization (n = 5; 23%; 95% CI [0.08 – 0.45]), strange or remarkable bodily sensations (n = 4; 18%; 95% CI [0.05 – 0.40]), and visual hallucinations (n = 2; 9%; 95% CI [0.01 – 0.29]).” (page 7, lines 184 – 193). 

 [Reviewer #2] 

[Authors] We would like to thank reviewer #2 for this thorough review. We are grateful for all the valuable remarks and recommendations for additional recent literature which has helped us to improve our manuscript. We will address all the comments point-by-point below. The mentioned pages and lines refer to the highlighted manuscript file.

1. [Reviewer #2] “Interviews were analysed with directed content analysis, using an existing framework [1].” Please name the framework. 

[Authors] We referred to the authors of the framework in the abstract: “…, using an existing framework by Lindahl et al. [1].” (page 2, line 39).

2. [Reviewer #2] “The seven existing domains of AEs were confirmed: cognitive, perceptual, affective, somatic, conative, sense of self, and social.” Take out “the”, and I suggest changing language from “confirmed” to “observed”, see below.

[Authors] We took out the word “the” throughout the whole manuscript and changed the word “confirmed” to observed throughout the whole manuscript.

3. [Reviewer #2] “Mindfulness-Based Interventions (MBIs) are widely used in both clinical and non-clinical populations, and although the benefits of MBIs in psychiatric populations are well documented [4], the same applies to MBIs.” Grammatically confusing. What is “the same” that applies to MBIs?

[Authors]. Our apologies for this mistake. We changed the sentence to: “Mindfulness-Based Interventions (MBIs) are widely used in both clinical and non-clinical populations, and the benefits of MBIs in psychiatric populations are well documented (1). However, remarkably little attention has been given to potential (S)AEs following MBIs (2), even though the occurrence of (S)AEs following meditation are not uncommon.” (page 3, lines 54-57). 

4. [Reviewer #2] As mentioned earlier, Britton, et al. 2021 should be cited, as that article outlines a methodological approach and has described base rates for AEs. Also Goldberg, et al. 2021 will be relevant.

[Authors] We thank the reviewer for mentioning these articles, as indeed these were published after submission of the current manuscript. We read the articles closely and have brought the manuscript up-to-date with the more recent literature. 

5. [Reviewer #2] I suggest caution in generalizing Lindahl, et al.’s study to MBIs, as that research was done with long-term Buddhist meditators. This article mentions this as an aside at the end, but it should be clear from the outset.

[Authors] We agree with the reviewer that this should be mentioned at the beginning of the article. We included the following sentences in the introduction section at page 3, lines 66-69: “It is, however, an open question whether these results are generalizable to MBIs (3). MBIs use a structured format, and a gradual increase in the length of meditation exercises. They include psychoeducation and inquiry, thereby allowing exchange of experiences, which might be important to prevent (S)AEs (3).”

6. [Reviewer #2] The authors mention that this research relied on an adapted version of MBCT. Reasons and implications of adaptation should be stated (perhaps in discussion is OK).

[Authors] We made some adaptations to the original protocol to tailor the intervention to the BD population. Adaptations were based on qualitative feedback of participants in two pilot groups. We added the reasons for adaptation to the methods section: “Some adaptions to the original protocol were made in order to tailor the intervention to BD. These adaptions included tailoring psychoeducation to BD, adding a partner session (session six), and instructing the teacher to add more movement exercises and repeatedly bringing focus to self-care (4).” (page 4, lines 101-105).

7. [Reviewer #2] Weekly completion of questionnaires is a definite strength. However, if only 12 AEs that were a-priori deemed most relevant to BD were indicated, this may also leave important AEs out. AEs may have as much to do with the intervention as with the population, so making the items only focused on the specifics of the population may miss important characteristics. By way of analogy, an (S)AE of aspirin is Reye’s syndrome. However, if probing AEs of aspirin among low back pain sufferers, only asking about low back-relevant AEs would miss this serious AE. To be clear, the current questions seem appropriate and—if selecting 12—the items used appear to be good choices even if they do leave out (for example) change in social relationships.

[Authors] We agree with the reviewer that by selecting possible (S)AEs that only focus on the population, we might have missed other (S)AEs relevant to the intervention. However, we carefully composed the questionnaire ensuring that all main dimensions from the study of Lindahl et al. were represented. We noticed that we did not state this in our methods clearly enough. On page 4, line 111-116 we added: “Three psychiatrists with expertise in BD (RK, ER, and MLvB) and three clinical researchers in mindfulness (AS, MH and IH) discussed the 59 categories of meditation-related effects from the study of Lindahl et al. [1] and selected 12 most relevant to patients with BD to include in the self-report questionnaire (Supplementary Table 1). When selecting these 12 items, it was ensured that all main categories of Lindahl et al. [1] were represented.” 

We included an open-ended question were patients were able to report any other (S)AEs that they deemed important to mention. 

8. [Reviewer #2] Small concern: IH knew the patients and was responsible for recruiting them. Was there any concern that patients may under-report AEs to the person who recruited them, especially if she is a junior researcher? 

[Authors] We do not expect that the prior relationship between IH and the patients was of influence on the reporting of AEs, although we do not know this for sure. The prevalence of AEs was based on the self-report questionnaires which patients received in their workbook and handed in to their teacher, so IH was not involved in that. During the interview the interviewers emphasized that there are no right or wrong answers, and that patients should mention anything that they deemed important. Both interviewers probed for more information when necessary and during all of the interviews they asked patients whether they experienced other AEs that were not mentioned (either in the weekly questionnaire or during the interview), giving the patients multiple opportunities to discuss these experiences. 

9. [Reviewer #2] The authors discuss the construal of AEs as ultimately positive in many cases. For reference in the varied interpretation of meditation-related AEs, and some of the context around positive vs. negative interpretation of AEs as “progress”-related, they may also find a recent paper by Lindahl, et al. (2020) relevant.

[Authors] We thank the reviewer for the recommended literature. We do recognize that we, as clinicians and researchers, have a more positive interpretation of AEs when patients progressed or learned from it, then when they didn’t. However, we have to mention it were not the authors who interpreted the AEs as positive or negative. During the interview, patients were asked how they interpreted the AEs when looking back at the experience. The patients themselves interpreted the AEs as positive when they had learned something from it. 

10. [Reviewer #2] Was any kind of intent to treat approach used with drop-outs? Was there any indication of why people dropped out and whether this may have been related to AEs? Also, it’s not clear to what extent drop-outs are taken into account in the current analyses. A problem sentence is: “Of these, 29 (50%) patients reported one or more AEs and were invited for a semi-structured interview…”. “of these” is confusing, since prior to this sentence a general sample is listed followed by number drop-outs. So is this referring to all participants prior to drop-out, or after excluding drop-outs?

[Authors] Thank you for noting. We kindly refer to Fig 1, the participant flowchart, where reasons for dropping out are included. Three patients indicated reasons for dropping out that might have been related to AEs (i.e. mania and depression). These are mentioned under ‘Prevalence and course of Adverse Effects’ (page 6). We did not receive weekly questionnaires from any of the drop-outs, so we have no further information regarding possible AEs. As we did not receive weekly questionnaires from the drop-outs, they were excluded from further analysis and the prevalence is based on the patients who did hand in weekly questionnaires (29 out of 58). To make this more clear in text, we added the following sentence: “None of the patients who dropped-out handed in their self-report questionnaires, and therefore were excluded from further analyses” (page 6, line 168-170). Furthermore, we replaced the sentence starting with “of these” with “of the 58 patients who handed in their self-report questionnaires, 29 (50%)….” (page 6, line 171). 

11. [Reviewer #2] The description of qualitative analyses is quite sparse. Are there any data on initial rates of agreement between coders? What was the “existing framework” for directed content analysis? Etc.

[Authors] Unfortunately, we have no data on the initial agreement rates between coders. We elaborated on the qualitative analyses by adding the following sentence in the methods section on page 6, lines 153-155: “We used the framework by Lindahl et al. (5) as a basis to identify coding categories and coded our transcripts using the predetermined codes from their codebook, which is available online (5).”

12. [Reviewer #2] Was chronicity/duration of AEs monitored? If so, how? The authors indicate later in the paper that enduring AEs were not observed, but it is not clear how they are able to make such a statement, especially if AEs were not monitored after end of trial.

[Authors] This is a good point. This depends on how we define long-term. It is true that not all AEs disappeared at the end of the study. Some patients still experienced AEs during meditation. However, patients described that they knew how to handle these AEs, as a result of which they did not have a long-term impact on daily functioning. Indeed, as the reviewer observed, there is a long-term effect but not a long-term impact. In line with the study of Britton et al., 2021, we decided to change long-term effect to lasting bad effect. 

“Furthermore, we did not find any lasting bad effects, which is in contrast with Britton et al. (6) who found that 15% of participants still experienced lasting bad effects for longer than one week after the (S)AEs following MBCT had occurred. One explanation for this difference might be that this population of patients with BD had a long treatment history in which they had possibly become familiar with adverse emotional experiences such as panic attacks, re-experiencing of traumatic memories, depersonalization, and with ways to cope with them. Furthermore, by actively monitoring (S)AEs is it possible that patients were inclined to share these challenging experiences sooner, after which the teacher was able to provide them with tools to mitigate these (S)AEs.” (page 13, line 373-382).

13. [Reviewer #2] “In the current study, 41 of the 59 categories of the phenomenology codebook by Lindahl et al. [1] were confirmed (Supplementary Table 3).” I suggest stating “observed” instead of “confirmed”, since the design of the study is not well suited to confirm or disconfirm these categories (i.e., the authors did not produce an independent coding system and find it to accord with the Lindahl, et al. themes, nor did they sample an adequate number of participants (22 vs. approximately 60 in the Lindahl et al. study). To be clear this is not a problem with the study or its methods, which are sound. Rather, this is just about what the study is and isn’t positioned to do. I therefore suggest “observed.”

[Authors] We agree with the reviewer that they methodology of this study is not suited to confirm any of the domains of Lindahl et al. Therefore, we changed it to observed throughout the whole manuscript.

14. [Reviewer #2] “Both the number of patients who reported AEs and the average number of different AEs per patient declined over the course of MBCT, from 12 (21%) in week 1 to 8 (14%) in week 7 and from 3.6 in week 1 to 1.5 in week 7, respectively (see Fig 3a and Fig 3b).” This is a bit confusing – are these total patients or averaged numbers of AEs? I am assuming that it’s patients followed by averages, but please make this clear.

[Authors] These are indeed total number of patients and average number of AEs per patient. We tried to clarify this in text by the following sentence: “The total number of patients who reported AEs per week declined over the course of MBCT, from 12 (21%) in week 1 to 8 (14%) in week 7 (Fig. 3a). In addition, the average number of reported AEs per patient per week declined from 3.6 in week 1 to 1.5 in week 7 (Fig. 3b).” (Page 7, lines 196-198). 

15. [Reviewer #2] “Changes in doubt, faith, trust, or commitment were frequently reported as well (see Supplementary Table 3) , which referred to insecurities about patients’ ability to learn mindfulness skills and retain a stable a mood, and doubts about whether mindfulness could be harmful in the light of the occurring AEs”. What is “frequently”?

[Authors] We kindly refer to Supplementary Table 3, which depicts that the phenomenology codebook of Lindahl et al. including the number of patients in the present study who reported these separate codes. Changes in doubt, faith, trust or commitment were mentioned by 9 out of 19 (47%) patients. 

In order to make this more clear, we changed the sentence to: “Changes in doubt, faith, trust, or commitment were reported by almost half or the patients, which referred to insecurities about patients’ ability to learn mindfulness skills and retain a stable a mood, and doubts about whether mindfulness could be harmful in the light of the occurring AEs.” (page 9, lines 225-229).

16. [Reviewer #2] “As no SAEs were found in this study, the text below will only mention AEs.” The authors do not explain how “seriousness” was determined, a very important omission to correct. This explanation would naturally fall in the “monitoring serious adverse effects” section, but that section only includes the questionnaire and interview approach.

[Authors] We thank the reviewer for this important remark. We used the definition of the FDA to indicate seriousness. We included the definition of AEs and SAEs that we used in the methods section. At page 5, lines 116-121 we added: “AEs refer to any meditation related-effects that occurred during the course of MBCT, which patients indicated as having a negative valence or negative impact on daily functioning (6). The experience was called an SAE when the outcome was death, life-threatening, hospitalization, disability, or permanent damage to conduct normal life functions or quality of life, or when it required treatment to prevent the above (7).”

17. [Reviewer #2] Please define each of the 4 new categories for influencing factors. I.e., the authors write “Patients mentioned several factors that existed before the start of MBCT and that might predispose to the occurrence of AEs.” Are these factors deemed by the patients, or by the researchers, to be predisposing? And if by the researchers, based on what criteria? How were these predisposing factors (i.e., personality traits) identified – were they spontaneously reported by participants, or queried by interviewers?

[Authors] This is an important question. The different categories were identified by the researchers and were based on psychiatric case formulation as used in clinical practice. We used the following criteria for each of the categories, as mentioned in the manuscript as well: predisposing refers to any factors that were already apparent before start of MBCT (this includes for example the personality traits, traumatizing experiences in the past, etc.) and were identified as risk factors by patients. Precipitating factors refer to any factor that occurred during the course of MBCT that might have had an influence on the occurrence of (S)AEs (such as the meditation practice itself). Perpetuating and mitigating factors refer to specific actions/factors in response to the (S)AEs that were considered helpful or unhelpful.

The categories were not specified during the interviews, but the interviewers did query about factors that might have had an influence on the occurrence of (S)AEs (see Supplementary Table 2). The patients then spontaneously reported factors that they thought had an influence on the occurrence of the (S)AEs. For example, patients who re-experienced traumatic memories indicated that traumatic experiences in the past influenced the occurrence of this AE. We ensured to include this information in the methods section. On page 5, lines 131-140 we added: “The following themes were discussed: 1) occurrence of AEs; 2) responses to AEs; and 3) interpretation of AEs. Supplementary Table 2 provides an overview of the open-ended questions that were used during the interview. The interviews started with the question: “What kind of adverse or unexpected experience(s) did you have that you considered to be related to MBCT?” After that, the three above mentioned themes were discussed per AE that patients had experienced. The interviewers used open-ended questions only, factors were not specified beforehand and were therefore always reported spontaneously by patients. At the end of the interviews, when patients indicated that all AEs were discussed, the interviewers screened the self-report questionnaires to check whether that was indeed the case. ”

18. [Reviewer #2] Please explain the categorization of “trying to continue the meditation exercises despite the occurrence of AEs” as a maladaptive (automatic) coping strategy, given that instructions to “meditate through it” are a conspicuous concern in popular meditation instruction. If this was in line with existing instruction and literature on meditation, calling it “automatic” localizes this issue to the patient, rather than placing appropriate responsibility with instructors (perhaps not those in the present study) who repeat this instruction.

[Authors] This is an important remark. We are aware that the language used by the instructors is of great importance when looking at (S)AEs. In de current study, patients also mentioned that these instructions were sometimes maladaptive (see for example page 11, line 289), as a consequence of which we included in the discussion section of the paper : “teachers should be aware of situations in which ‘just staying with the experience’ might not be the best instruction”) (page 15, lines 419-421). We coded this with relationship to teacher. The maladaptive (automatic) coping strategies do refer to coping strategies that patients applied themselves. For example, a patient who experienced panic during meditation mentioned that she continued with the meditation practice because she always “pushes through because giving up is not in my nature”. 

We tried to specify this difference in text. At page 11, line 287 we added: “Some maladaptive (automatic) coping strategies of patients themselves, such as….” and at line 291: “Some specific unhelpful reactions of the teacher were mentioned, such as suggesting that they should “stay with the experience” or…”. 

19. [Reviewer #2] “Mindfulness skills” as a mitigating factor is not an immediately clear category – please provide a definition/explication, including how the category was generated. Were these described as “mindfulness skills” by participants, or categorized as such by the researchers? How are these different from “ground activities” (which include focusing on the breath) for instance? 

[Authors] We agree that this distinction is not entirely clear. The patients mentioned the specific factors in response to the question: “how did you respond to these experiences” and “which actions/strategies did you find particularly helpful?” Afterwards, the mentioned factors were coded as grounding activities or mindfulness skills by the researchers. 

With grounding activities we refer to actual behavioral responses that patients performed in order to anchor in the present moment (e.g. touching something, opening eyes, redirecting attention to breathing, etc.). With mindfulness skills we refer to responses on an emotional / cognitive level (incl. allowing/accepting, decentering and soothing). However, we agree that both these domains fall under the umbrella of “mindfulness skills”. Therefore we decided to relocate the codes under the category grounding to mindfulness skills, as all these approaches are directed at alleviating the intensity of occurred (S)AEs. 

20. [Reviewer #2] It will be important to resolve the following concern: The article initially states that most patients interpreted the AEs to be ultimately therapeutic or positive. However, the Consequences section begins by writing “First of all, some patients believed that the AEs would persist or even worsen over time and tended to avoid practicing mindfulness altogether in order to prevent this.” Or at the end of that paragraph, “Some patients experienced AEs as merely having negative impact, because they did not feel that they had learned anything from it.” Given the small number of participants, this is non-trivial and would seem to indicate that the earlier statement be revised. One possible revision would be to add something like “although some only experienced AEs as negate ve.” With monitoring adverse effects, it is important not to elide the experiences of those patients. It is also important to bear in mind that the full extent of distress may not be observed within the time window of the study. For example, when comparing AEs that did vs. did not reduce over time (Supplementary Table 4), the number that do not decrease outnumber those that did. Likewise, the “majority” who interpreted the AEs as part of a growth process was 11 vs. 8 who did not, which I is a VERY SLIM majority. Since an AE like depersonalization can seem fine within the time frame of an 8-week MBCT treatment but, without improvement, can become highly distressing after 3 months, it is inappropriate to characterize these as mostly positive overall.

[Authors] We thank the reviewer for addressing this important point, and we agree that the conclusion of the manuscript was overly focused on those patients who experienced the AEs as part of a growth process. We feel that the sentence that the reviewer proposed to add to the conclusion “…, although some patients only experiences AEs as negative” suits well with the overall conclusion of the manuscript, so we added this sentence to the abstract (page 2, line 50). Furthermore, we changed ‘majority’ to ‘more than half’ throughout the manuscript. 

21. [Reviewer #2] The Consequences section again mentions specific “mindfulness skills” which makes me wonder whether these were queried in the interview, or whether they were ultimately categorized as such by the researchers. 

[Authors] As mentioned at remark #19, we relocated the grounding to mindfulness skills. 

22. [Reviewer #2] Given the spread of those who did and did not find AEs to ultimately be beneficial, it might be useful to see the numbers behind these differences.

[Authors] We included the number of patients who did report a benefit from the AEs (n = 11 (58%) patients) and those who did not report beneficial effects (n = 8 (42%)) in the consequences section (page 12). 

23. [Reviewer #2] “During the course of MBCT patients gradually learn to develop acceptance and self-kindness, which have been shown to reduce psychological distress [10, 15, 16]. This might explain why the number of AEs reported by patients starts to decline after three weeks. These results suggest that allowing difficult or adverse emotions can result in a reduction of associated fears and habituation. This is in accordance with exposure-based therapies, where a temporary increase of anxiety is considered a necessary vehicle to change patients’ beliefs and emotions [7]” This may be overstating the case, and I would suggest caution in making such an interpretation given the kind of—and amount of—data here. For instance, an alternative explanation is that, as with many adverse experiences, impacts are shorter-term, likelier to occur in tandem with any novel experience, and that they recover on their own because people are by and large resilient. Decline in reporting could be due to Hawthorne effects, as patients who report an AE 3 weeks in a row may be reluctant/tired/habituated to it on week 4 even though they still experience it. It is thus a strength of the study that these patients were subsequently interviewed (and yet the interviews did not suggest that decrease was the majority experience). Finally, research suggests that early losses in treatment are not good predictors of longer-term treatment gains (e.g. “it will get worse before it gets better”)—this is a common assumption but is not well supported (Flückiger et al., 2013; Koffmann, 2018, 2020; Lutz et al., 2013).

[Authors] We agree that there are multiple explanations for the finding that the number of reported AEs declines during the course of MBCT. As patients indicate that specific mindfulness skills, including grounding, accepting, allowing, and soothing, are helpful in mitigating AEs, and given the fact that over half of patients indicate that the AEs were part of a therapeutic process, we do feel that this explanation should be mentioned in the manuscript. However, we decided to remove the sentences “These results suggest that allowing difficult or adverse emotions can result in a reduction of associated fears and habituation. This is in accordance with exposure-based therapies, where a temporary increase of anxiety is considered a necessary vehicle to change patients’ beliefs and emotions” as this might indeed be overstating the case. 

We also added the alternative explanation put forward by the reviewer to the manuscript at page 13, lines 362-364: “However, another explanation of this decline may be that even though patients still experienced AEs during the course of MBCT, they may have become reluctant to report this on the self-report questionnaires.”

24. [Reviewer #2] “In the current study, with a vulnerable population of patients with BD, MBCT seemed reasonably safe.” This is a value judgment. See paper by Britton, et al. (2021) for discussing safety.

[Authors] We agree with the reviewer and decided to remove the following sentence from the manuscript: “In the current study, with a vulnerable population of patients with BD, MBCT seemed reasonably safe.” and instead add the following sentence: “Although the occurrence of AEs is not rare, even in this population with severe mental illness, we did not find any SAEs” (page 13, line 369).

25. [Reviewer #2] Given that one patient died by suicide (a low base rate event), I am concerned about the authors’ justification in saying they did not observe suicidality as an AE among patients.

[Authors] We understand the reviewers’ concern about this point, and we would like to take this opportunity to explain how we came to this conclusion. 

After the suicide was reported to the leading researcher, we immediately started an investigation of all relevant research and clinical data available to evaluate whether the intervention might have had any possible causal link with this event, also as a part of the obligatory report to the ethical committee. According to the psychiatrist and psychiatric nurse of the patient, his depression was precipitated by impending life-events: a divorce from his wife, having to sell his house and live with his mother. Pharmacological treatment had not relieved the depression, which increased the despair of the patient. Eventually, five months after the completion of the MBCT, the patient died of suicide. The psychiatrist, psychiatric nurse, and the ethical committee concluded that they did not think there was a connection between the suicide and MBCT. 

We included the following sentence in the manuscript in order to prevent confusion on this by future readers: “– the ethical committee considered this event unrelated to MBCT” (page 6, line 174). 

26. [Reviewer #2] Given the low base rates of harms in general, it is not justified based on lack of observation in this trial to say that these do not occur. 

[Authors] We agree with the reviewer, and we certainly did not mean to imply this. We are not entirely sure to which sentence the reviewer refers, but we suspect that this has to do with the following sentence in the discussion section: “In the current study, with a vulnerable population of patients with BD, MBCT seemed reasonably safe.” (page 11, line 302). As mention in a previous comment, we deleted this sentence to make sure the readers will not think we are implying that MBCT goes without harm.

Furthermore, we already mentioned in our limitations that the sample size of the current study might be too small to identify SAEs: “Another possible limitation of the study is the sample size, which might not be large enough to identify (S)AEs that occur less frequently (8)” (page 15, line 442-444)

27. [Reviewer #2] “Both precipitating and perpetuating factors provide insight into ways to improve MBIs, for example by managing expectations and inform patients about the possibility that symptoms may initially get worse” See earlier comment that “things get worse before they get better” is not well borne out in psychotherapy, and that early losses do not predict later gains.

[Authors] In line with the articles that the reviewer provided on this topic, we decided to reframe this sentence to the following: “Both precipitating and perpetuating factors provide insight into ways to improve MBIs, for example by managing expectations and inform patients about the possibility that they might become more aware of difficult emotions or that unexpected, negative effects might occur.” (page 14, lines 408-411). 

28. [Reviewer #2] “Informing patients in advance may prevent them from dropping out due to these AEs.” Why is it inappropriate to drop out due to AEs? This seems like it would be a reasonable strategy with many other therapies. It also stands in contrast with the researchers’ observations that persisting with the practice is a potential exacerbating factor. I suggest changing this language to indicate that patients should received informed guidance about their options to drop out, switch therapies, etc. in the event of AEs.

[Authors] We agree that patients should be able to drop out or switch therapies in the event of AEs, and to do this, ideally, in shared decision making with the teacher. Therefore, we changed the sentence to the following: “Informing patients in advance may help to come to a balanced and ideally shared decision with the mindfulness teacher on whether to stay with the practice (with perhaps some adaptations), or to consider other treatment alternatives.” (page 14, lines 411-413). 

29. [Reviewer #2] The authors state that long-term AEs were not observed, but since all AEs did not disappear at end of study, how can they know?

[Authors] See our earlier comment. We changed long-term to lasting bad effect in line with the study of Britton et al. 2021. 

“Furthermore, we did not find any lasting bad effects, which is in contrast with Britton et al. (6) who found that 15% of participants still experienced lasting bad effects for longer than one week after the (S)AEs following MBCT had occurred. One explanation for this difference might be that this population of patients with BD had a long treatment history in which they had possibly become familiar with adverse emotional experiences such as panic attacks, re-experiencing of traumatic memories, depersonalization, and with ways to cope with them. Furthermore, by actively monitoring (S)AEs is it possible that patients were inclined to share these challenging experiences sooner, after which the teacher was able to provide them with handles to mitigate these (S)AEs.” (page 13, lines 373-382).

30. [Reviewer #2] Not sure why the patterns described in strengths and limitations are deemed “over”-reporting. It seems appropriate to query AEs and have a system for determining whether these were harmful, but the methods are well designed to correct for any over-estimation due to the sensitivity of the instruments.

[Authors] We understand the reviewers’ point. However, we did find some overreporting in our study in the weekly self-report questionnaires. As mentioned in the results section, some patients indicated they had experienced (S)AEs related to MBCT in their self-report questionnaires, but during the interview it became apparent that they were actually mistaken. For example, one patient reported pain as an AE, but indicated in the interview this was due to a leg fracture and had nothing to do with the MBCT. Accordingly, one patient who reported sweating as an AE, but said in the interview this was due to her menopause. Excluding these errors, the prevalence of AEs changed from 50% to 38% . Therefore, we do think it is important to mention this as a limitation and to emphasize that the adverse events are correlated to meditation/MBCT. 

31. [Reviewer #2] A limitation is that drops due to AEs were not recorded.

[Authors] We agree that it is a limitation that we did not interview the drop-outs, as we have no information on whether they might have experienced (S)AEs. This was already included in the discussion section: “Patients who dropped out of MBCT or did not hand in the self-report questionnaires were not approached, while SAEs might be overrepresented among them.” (page 15, line 438-440). 

References

1. Goldberg SB, Tucker RP, Greene PA, Davidson RJ, Wampold BE, Kearney DJ, et al. Mindfulness-based interventions for psychiatric disorders: a systematic review and meta-analysis. Clinical Psychology Review. 2018;59:52-60.

2. Wong SY, Chan JY, Zhang D, Lee EK, Tsoi KK. The safety of mindfulness-based interventions: a systematic review of randomized controlled trials. Mindfulness. 2018;9(5):1344-57.

3. Baer R, Crane C, Miller E, Kuyken W. Doing no harm in mindfulness-based programs: conceptual issues and empirical findings. Clinical psychology review. 2019;71:101-14.

4. Hanssen I, Huijbers MJ, Lochmann-van Bennekom M, Regeer E, Stevens A, Evers S, et al. Study protocol of a multicenter randomized controlled trial of mindfulness-based cognitive therapy and treatment as usual in bipolar disorder. BMC psychiatry. 2019;19(130):1-10.

5. Lindahl JR, Fisher NE, Cooper DJ, Rosen RK, Britton WB. The varieties of contemplative experience: A mixed-methods study of meditation-related challenges in Western Buddhists. PloS one. 2017;12(5):e0176239.

6. Britton WB, Lindahl JR, Cooper DJ, Canby NK, Palitsky R. Defining and measuring meditation-related adverse effects in mindfulness-based programs. Clinical Psychological Science. 2021:2167702621996340.

7. FDA. What is a serious adverse event? 2016 [Available from: https://www.fda.gov/safety/reporting-serious-problems-fda/what-serious-adverse-event.

8. European Medicines Agency. Note for guidance on statistical principles for clinical trials. In: Agency EM, editor. London2006.

---

## [Decision Letter · Decision Letter 1]

24 Sep 2021

PONE-D-21-18959R1Adverse or therapeutic? A mixed-methods study investigating adverse effects of Mindfulness-Based Cognitive Therapy in bipolar disorderPLOS ONE

Dear Dr. Hanssen,

Thank you for submitting your manuscript to PLOS ONE. After careful consideration, we feel that it has merit but does not fully meet PLOS ONE’s publication criteria as it currently stands. Therefore, we invite you to submit a revised version of the manuscript that addresses the points raised during the review process. As you will see, both peer reviewers appreciated your revisions and believe the manuscript is much improved. Reviewer #1 simply asks for a little bit more transparency and commentary about the methods/analytic approach. I expect that this revision will be easily addressable.

We look forward to receiving your revised manuscript.

Kind regards,

Ethan Moitra

Academic Editor

PLOS ONE

Journal Requirements:

Reviewers' comments:

Reviewer's Responses to Questions

**Comments to the Author**

1. If the authors have adequately addressed your comments raised in a previous round of review and you feel that this manuscript is now acceptable for publication, you may indicate that here to bypass the “Comments to the Author” section, enter your conflict of interest statement in the “Confidential to Editor” section, and submit your "Accept" recommendation.

Reviewer #1: (No Response)

Reviewer #2: All comments have been addressed

2. Is the manuscript technically sound, and do the data support the conclusions?

Reviewer #1: Yes

Reviewer #2: Yes

3. Has the statistical analysis been performed appropriately and rigorously? 

Reviewer #1: (No Response)

Reviewer #2: Yes

4. Have the authors made all data underlying the findings in their manuscript fully available?

Reviewer #1: Yes

Reviewer #2: No

5. Is the manuscript presented in an intelligible fashion and written in standard English?

Reviewer #1: Yes

Reviewer #2: Yes

6. Review Comments to the Author

Reviewer #1: I am generally happy with the authors' responses to my previous few comments. However, I still have issues with the methods section. I did mention a logistic regression for example before and the authors responded that this is presented in the results section. Perhaps i should have rephrased my point original point, but, from my point of view, if it's not described in the methods section, it did not happen. And the analysis plan for the quants is a single sentence that does not mention such an analysis. So I would urge the authors to expand the data analysis section to include all the analyses they conducted, clearly.

Reviewer #2: I would like to commend the improvements and serious consideration that the authors have given to this article and its revision. I am supportive of the changes that have been made overall, and would like to recommend acceptance.

7. PLOS authors have the option to publish the peer review history of their article (what does this mean?). If published, this will include your full peer review and any attached files.

Reviewer #1: No

Reviewer #2: No

---

## [Author Response · Author response to Decision Letter 1]

6 Oct 2021

Reviewer #1: I am generally happy with the authors' responses to my previous few comments. However, I still have issues with the methods section. I did mention a logistic regression for example before and the authors responded that this is presented in the results section. Perhaps i should have rephrased my point original point, but, from my point of view, if it's not described in the methods section, it did not happen. And the analysis plan for the quants is a single sentence that does not mention such an analysis. So I would urge the authors to expand the data analysis section to include all the analyses they conducted, clearly.

[Authors] We thank the reviewer for the attentiveness regarding the methodology of this article, as it is important that this is as transparent as possible. 

The previous comment the reviewer refers to is as follows: “Characteristics associated with SAEs are not examined, in say a multiple logistic regression (or even a zero-inflated Poisson regression to capture multiple SAEs).”

The authors responded with the following: “We did explore whether baseline characteristics (e.g. age, gender, co-morbid psychiatric disorders, number of previous mood episodes, duration of illness, baseline depression, etc.) differed between patients who did and did not report (S)AEs. We did not find any differences, except for baseline anxiety (see page 7, line 178). This is an important finding, and we elaborated on this in the discussions section (page 14, line 402-407).”

We want to apologize for not addressing the reviewers’ question sufficiently. We did not conduct a multiple logistic regression to investigate which characteristics are associated with SAEs. We did however explore if there were any differences between the two groups (did or did not experience AEs) regarding these characteristics by means of chi-square and independent t-tests, as mentioned in the methods section (page 6, lines 147-150): “Quantitative data were analysed using SPSS version 25.0 [15], conducting chi-square and independent t-test statistics to compare demographic and clinical variables between patients who did and did not report (S)AEs.” As there was only one difference between those groups (baseline anxiety), we think that conducting a multiple logistics regression analysis will not be of added value. Moreover, our sample population might be too small for making a multivariate model. We hope the reviewer agrees with our decision to only look for differences between groups by means of chi-square and independent t-test statistics, and not by conducting a multivariate logistic regression.

Reviewer #2: I would like to commend the improvements and serious consideration that the authors have given to this article and its revision. I am supportive of the changes that have been made overall, and would like to recommend acceptance.

[Authors] We thank the reviewer for the helpful comments which helped us to improve our manuscript.

---

## [Editor Report · Decision Letter 2]

14 Oct 2021

Adverse or therapeutic? A mixed-methods study investigating adverse effects of Mindfulness-Based Cognitive Therapy in bipolar disorder

PONE-D-21-18959R2

Dear Dr. Hanssen,

We’re pleased to inform you that your manuscript has been judged scientifically suitable for publication and will be formally accepted for publication once it meets all outstanding technical requirements. I appreciate your attention that final reviewer comment. Your justification for the analyses you used is clear and I sufficiently articulated in this revision.

Kind regards,

Ethan Moitra

Academic Editor

PLOS ONE
---

## [Editor Report · Acceptance letter]

26 Oct 2021

PONE-D-21-18959R2 

Adverse or therapeutic? A mixed-methods study investigating adverse effects of Mindfulness-Based Cognitive Therapy in bipolar disorder 

Dear Dr. Hanssen:

I'm pleased to inform you that your manuscript has been deemed suitable for publication in PLOS ONE. Congratulations! Your manuscript is now with our production department. 

Kind regards, 

on behalf of

Dr. Ethan Moitra 

Academic Editor

PLOS ONE